# Reconstruction of the Expansion of Siberian Larch into the Mountain Tundra in the Polar Urals in the 20th—Early 21st Centuries

Valery Fomin [1,2,*], Anna Mikhailovich [3], Dmitry Golikov [4] and Egor Agapitov [1]

[1] Institute of Forest and Natural Resource Management, Ural State Forest Engineering University, Sibirskiy Trakt, 37, 620100 Yekaterinburg, Russia; agapitovem@m.usfeu.ru

[2] Institute of Natural Sciences and Mathematics, Ural Federal University, 19 Mira Street, 620002 Yekaterinburg, Russia

[3] Institute of Physics and Technology, Ural Federal University, 19 Mira Street, 620002 Yekaterinburg, Russia; anna.mikhailovich@gmail.com

[4] Botanic Garden of the Ural Branch of Russian Academy of Sciences, 8 Marta Street, 202a, 620144 Yekaterinburg, Russia; mit2704@gmail.com

[*] Correspondence: fominvv@m.usfeu.ru or fomval@gmail.com

**Abstract:** This paper presents results of analyzing the second half of the 20th–early 21st century changes in lateral spatial structure of *Larix sibirica* Ledeb. population in the upper treeline ecotone located on the Rai-Iz massif (Polar Urals, Russia). The GIS layers characterizing distribution of Siberian larch trees and undergrowth together with their crowns was produced for a 7.32 square kilometer area based on aerial images recognition. Using statistical models, we assessed probabilities for assigning trees to age intervals of 1–10, 11–40, and 40+ years based on the average radius of tree crown projection. These maps and layer showing locations of trees that grew in the upper part of the ecotone, and died during the Little Ice Age, allow for assessing specifics of forest cover proliferation at different parts of upper treeline ecotone, and comparing current location of the trees with one from the past. The proposed method for probability-based recognition of Siberian larch tree generations in the upper treeline ecotone using average crown radius can be used to reconstruct time and spatial forest dynamics at the upper growth boundaries for time spans up to 100 years and more.

**Keywords:** trees; treeline ecotone; upward shift; *Larix sibirica* Ledeb.; Polar Urals

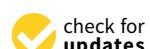

## 1. Introduction

Extensive research demonstrates the fact that forests advancing to the upper parts of mountain massif slopes located in different regions of Earth can be attributed to improvement of climate conditions that occurred in the 20th century and at the beginning of the 21st century [1–14]. Mountain systems subject for the studies of climate-driven forest cover dynamics at the upper boundaries of the forest are used as areas for monitoring biota reactions to global and local climate changes [2,13–17].

Intensive developments in the field of obtaining and processing high-spatial resolution satellite images, and ultra-high resolution images obtained with small unmanned aerial vehicles extended the spectrum of available solutions for obtaining and analyzing data on spatial position of the trees, which, in turn, create new opportunities for retrospective analysis of temporal and spatial dynamics of forests based on the quantitative assessment of age and specific biometrical parameters for the trees [17].

The manual method of tree recognition on the aerial images is still widely used due to the fact that automated methods for tree recognition do not allow for obtaining the required level of reliability when determining tree locations, and the size of tree crowns. Key reasons for this include nonuniform lighting conditions, depending on both time of day and shadows created by clouds, trees, and relief; the large variety of ground cover

due to combination of different factors including rock fragments, grass cover, and shrubs; and the specifics of larch tree growth (single-stem, multiple stem, and prostrate forms, and compact biological groups of the trees).

Key issues of the manual processing method include: (1) occurrence of optical illusions due to spatial context that affects object recognition, (2) subjectivity of operator assessments depending on personal image perception specifics, and (3) high labor intensity of recognition and data processing.

For example, color affects perception of object brightness [18], and the pattern on a uniform background seems to give more contrast compared to the case when the pattern is built within the same pattern of higher contrast (Chubb effect), at the same time changing spatial frequency of the image surrounding a pattern does not lower contrast of the former [19]. These examples demonstrate that it is necessary to focus on the method of operator-based object recognition in images. The goal of this work was to establish mapping models that characterize proliferation of Siberian larch trees into tundra and forested tundra in the second half of the 20th and early 21st centuries, based on data from deciphering images obtained using an unmanned aerial vehicle.

## 2. Materials and Methods

### 2.1. Study Area

The Study area with boundaries within the limits of 66°30′28″ N–66°47′42″ N, 65°49′28″ E–65°33′59″ E (Figure 1) is located on the southeastern macroslope of the Rai-Iz massif (Polar Urals, Russia). The massif is composed of ultrabasic rocks (peridotite). The Chernaya and Malaya Chernaya mountains, composed of gabbro, are located in the southeastern part of the massif (Figure 1b). Surface formation of the studied macroslope was affected by a glacier during course of the last global glaciation that ended about 10 Kya [20]. While moving, the glacier carried and mixed the abovementioned rocks; thus, the surface features both peridotite and gabbro. The study area surface features a large number of knolls, elongated ridges, and depressions. The hydrographic network includes the Yengaiu River, which forms the natural southern and western borders of the study area; several streams discharge into the Yengaiu River, together with a large number of temporary streams. The study area features multiple small lakes and reservoirs (Figure 1), some of which become dry in summer. The region of interest is located in the permafrost zone. The lake beds were formed by a glacier. Yareity Lake belongs to the glacier-dammed type. Our soil studies have shown that there is no permafrost in the root layer of trees (extending down to underlying rock). While performing soil research in the study area with hand tools and without heavy equipment, it is impossible to excavate to the depth of (presumable) permafrost. At the same time, the entire region and adjacent territories are located in the permafrost zone. Thus, we can presume the presence of permafrost in the study area, but cannot reach it.

Maximum elevation of the study area is 313 m above sea level. Pure Siberian larch (*Larix sibirica* Ledeb.) forest prevails in the upper treeline ecotone [2]. The lower part of the ecotone includes open larch forests, and closed forests with admixed Siberian spruce (*Picea obovata* Ledeb.) and mountain birch (*Betula tortuosa* Ledeb.). The undergrowth is composed mainly of dwarf birch (*Betula nana* L.), and Siberian juniper (*Juniperus sibirica* Bugsd.) [21,22]. The grass and shrub layer is composed mainly of blueberry (*Vaccinium uliginosum* L.), while true mosses and lichens prevail in ground cover [22].

### 2.2. Ground Measurements

Nine circular forest plots were established in the study area, located in three transects (three plots each) stretched along the elevation and tree density gradient (Figure 1c). For each plot with radius equal to 11 m, measurements were made to determine the location of each Siberian larch tree, sprout, or undergrowth exemplar. In order to do that, bearings (oriented from north) were taken at the center of each plot using an AR-1 boussole (Vologda Optical and Mechanical Plant, Vologda, Russia), and the distance to the trees was measured using a measuring tape. Geographical coordinates of sample plots were determined

using an Etrex 10 GPS receiver (Garmin Inc., Schaffhausen, Switzerland). The following biometrical characteristics were determined for each exemplar of Siberian larch: height, diameter at root collar, diameter at 1.3 m height (for trees with height exceeding this threshold), and maximum length of crown projection in two orthogonal directions. The height of the large trees was determined using Suunto clinometer (Suunto Inc., Finland, Vantaa), small trees were measured using a measurement tape. From the large trees, core samples were taken using a Haglöf increment borer (Haglöf Sweden AB, Långsele, Sweden).

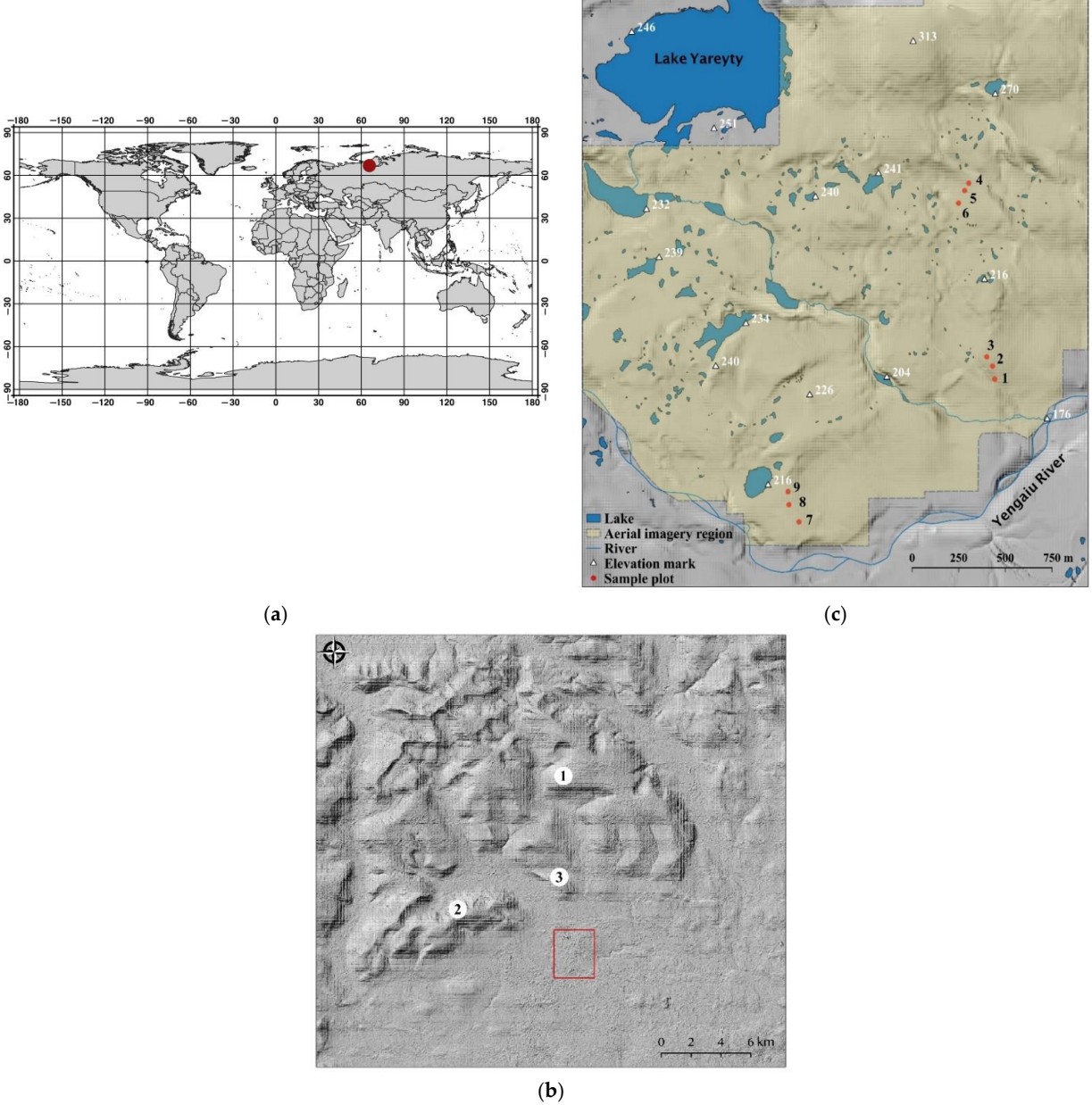

**Figure 1.** Study area location on the world map (**a**) and on the map of the Rai-Iz massif (**b**): 1—Rai-Iz massif, 2—Chernaya Mountain, and 3—Malaya Chernaya Mountain; shaded relief of the study region with sample plots, area of aerial imagery, hydrographic objects, and elevation benchmarks (**c**).

Several samples of saplings with different height were taken to determine the age of sprouts and undergrowth. Lintab dendrochronology complex (Rinntech Inc., Heidelberg, Germany) was used to determine tree age using core samples. Tree age was determined

using a method for coring height correction [23,24]. To perform that measurement, the age of the larch sapling with height corresponding with coring height was added to the tree age determined from the core sample.

Coordinates for every tree center were calculated in UTM projection using data collected during the field studies. The average crown radius was calculated based on maximum lengths of orthogonal crown projections and used to represent the tree crown in a model as a circle with equivalent radius (Figure 2).

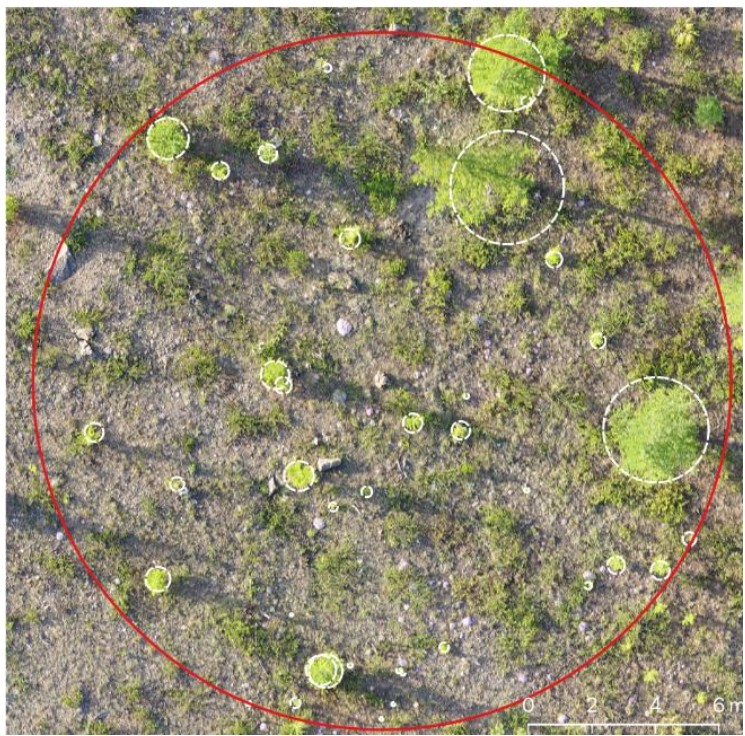

**Figure 2.** Fragment of image mosaics obtained using drone images, with superimposed polygonal layer of Siberian larch crowns, calculated using land-based measurements of orthogonal crown projections (white dash lines). The red line marks the boundary of a sample plot.

*2.3. Aerial Photography, Image Processing, and Tree Recognition*

The study area aerial photographs were taken from an altitude equal to 50 m using a Phantom 4 Advanced drone (DJI, Shenzhen, China). Aerial photography of the study area was performed in dry weather in July 2018, and, partially, in July 2019. At the time of UAV launch, a flight area was determined, usually with a rectangular shape. The UAV flight was performed automatically, under control of the PIX4Dcapture (https://www.pix4d.com/, accessed on 1 February 2022) mobile software application. Aerial photograph overlap within the flight line and among the neighboring lines was 80%. The Exiftool (https://exiftool.org/, accessed on 1 February 2022) application was used to examine the flight area. This application allows reading aerial image metadata, including the coordinates of the photograph center. Coordinates of image centers were used in the QGIS geographic information system (https://qgis.org/, accessed on 1 February 2022) to create a point layer for each flight area, which was in turn used to control execution of the flight plan. An orthophoto of a 7.32 sq. km study area was created using Metashape Professional software (Agisoft LLC, St. Petersburg, Russia) according to general workflow, described in the Agisoft Metashape User Manual [25].

Trees and undergrowth of Siberian larch were digitized on aerial images by hand in QGIS. Recognition of larch trees within the entire study area was performed after training by four operators using drone images from sample plots. Agreement of operator results was evaluated using compatibility analysis as per ISO 5725-2:2019 [26] standard. This analysis

includes calculation of two Mandel statistics: h—agreement of object recognition results obtained by different operators, and k—agreement of results obtained by an operator.

Forest plots were broken into sets training and testing sets. The first set included sample plot number 2, 3, and 7, while the training set included the remaining plots (i.e., plots number 1, 4, 5, 6, 8, and 9). Data from the training set was used to train operators, and results for Siberian larch recognition from the testing set were used to analyze the consistency of results obtained by the operators.

*2.4. Statistical Data Analysis*

The following method was used to place trees into three age intervals. All the data obtained during the course of direct measurements at forest plots were aggregated into a single file and ranged using the following age intervals: 1–10, 11–40, and >40 years of age. The first interval corresponded with placing a tree into the undergrowth category, and the upper boundary of the second interval was used to distinguish young trees from middle-aged ones. All trees, middle-age and above, were placed in the third category.

Data of direct measurements on forest plots underwent statistical processing using R software v. 4.1.2 (The R Foundation, Austria, Vienna) (https://www.r-project.org/, accessed on 5 March 2022), in particular, the MASS v. 7.3-55, caret v. 6.0-88, klaR, nortest v. 1.0-4, goft v. 1.3.6, dplyr v. 1.0.8, and yarrr v. 0.1.5 libraries. In course of R statistical software processing, parameters for average radius of crown projection distribution were evaluated using the fitdist function and graphical analysis (Q-Q plot, P-P-plot, CDF). The Weibull distribution provided the best fit for describing the distribution of values for average radius of maximum crown projection in two orthogonal directions within the limits of all three age intervals. For additional verification, a Weibull test was performed for every age interval using the Weibull test in goft library in R and confirmed correctness of distribution law selection. Table 1 contains parameters of Weibull distribution for the values of crown radius from all three age ranges.

**Table 1.** *p*-values for the test of verifying the attribution of crown radius values of Siberian larch trees by age intervals to Weibull distribution, and the Weibull distribution parameters for each of the intervals.

| Age Interval, Years of Age | | | | | | Model Parameter Relative Units |
|---|---|---|---|---|---|---|
| **41 and more** | | **11–40** | | **1–10** | | |
| **0.740** | | **0.252** | | **0.056** | | *p*-**Value** |
| 0.242 | 1.462 | 0.112 | 1.123 | 0.083 | 0.791 | shape |
| 0.240 | 1.560 | 0.027 | 0.400 | 0.028 | 0.155 | scale |

## 3. Results

*3.1. Recognition of Undergrowth and Trees of Siberian Larch*

Figure 2 features a fragment from image mosaics showing the study area in the region of forest plots with a superimposed vector layer of round templates approximating the crowns of Siberian larch undergrowth and trees, based on direct measurements performed on forest plots. Spatial superposition of the tree crown layer on the mosaic of areal images allows for assessing precision of mapping size and positions of the young and mature larch trees obtained by using both data sources. Results demonstrate good correspondence of locations and size for tree crowns determined using data from aerial images and forest plots.

In general, compatibility analysis proved conformance of operator-based assessments. Operators provided stable recognition of a similar number of trees within natural variability boundaries in the course of iterative experiments, and differences among the operators were negligible. Some specifics could be compensated with corrective actions, for example, by periodic retraining, in order to prevent future occurrence of out-of-limit values for the test statistics.

Resulting data demonstrate that the abovementioned parameters of the objects on image mosaics for study area allow recognition of undergrowth and mature tree samples of Siberian larch. The selected method, provided that operators are trained, allows efficient recognition of *Larix sibirica* Ledeb. trees growing in the upper tree line ecotone of the Polar Urals.

### 3.2. Assessment of the Probability of Three Generations of Siberian Larch Recognition

Figure 3 presents probability density graphs for the average tree crown projection radii of Siberian larch specimens located in forest plots for three age intervals: 1–10 (age category (I), 11–40 (II), and >40 years of age (III). Vertical lines signify intersection points for the curves, and break the range of radius values into three areas according to the probability of placing a larch specimen into one of three age ranges. For specimens with a crown projection radius close to 0.17 m, probability for placement into age interval of 1–10 years exceeds the probability of it being placed into other age ranges. For the trees with crown radii from 0.17 to 0.69 m from age category II, the probability of being placed into the 11–40 years of age range is higher than for other intervals. For age category III, a chance that tree age will be over 40 years is higher than for the other age intervals. These values of crown radii were used to place trees with crowns recognized on aerial images into age intervals.

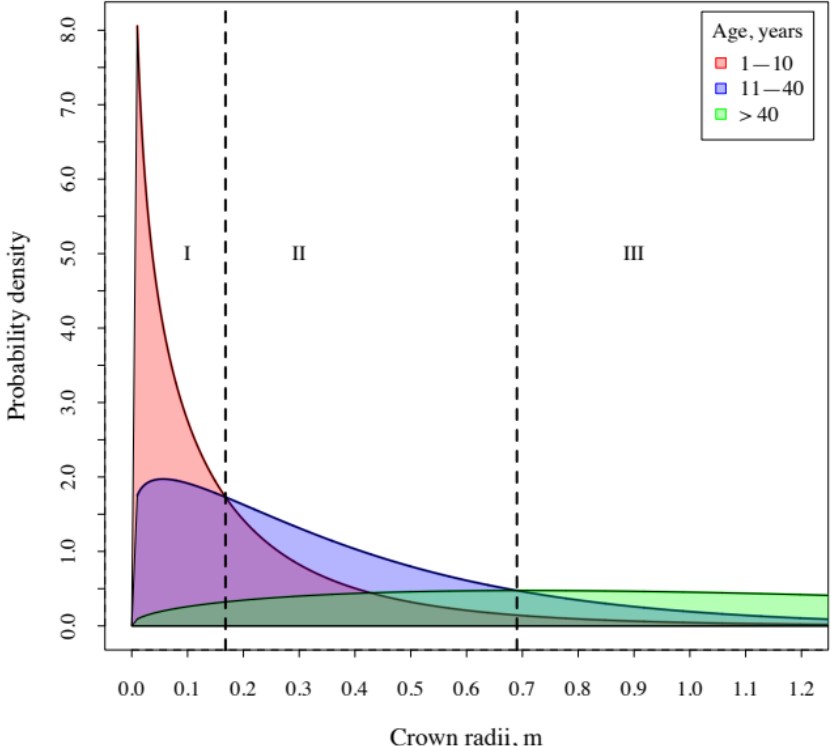

**Figure 3.** Probability density graphs for the values of average crown radius measured for Siberian larch specimens on sample plots for three age intervals. Vertical lines mark intersection points of the lines, breaking the range of radius values into three areas based on probability of a specimen falling into one of the age categories: 1–10 years (category I), 11–40 years (category II), and over 40 years (category III).

### 3.3. Mapping of Siberian Larch Trees and Tree Remnants

Figure 4 shows a location map of Siberian larch specimens (88,833 trees) in the upper treeline ecotone, created by recognition of drone aerial photos, and an enlarged fragment of the same map (Figure 4b) marked on Figure 4a with a red rectangle. The map shown on Figure 5a uses color dots to reflect the location of Siberian larch specimens belonging to different age categories, while Figure 5b shows location of the trees with age exceeding

40 years, and tree remnants (trees that were growing within the study area and died during the Little Ice Age). The method and results of tree remnant mapping is described in Fomin et al. [17]. Figure 6 shows the distribution map of Siberian larch for two age categories: 1–10 and 11–40 years, with specified locations of the tree remnants.

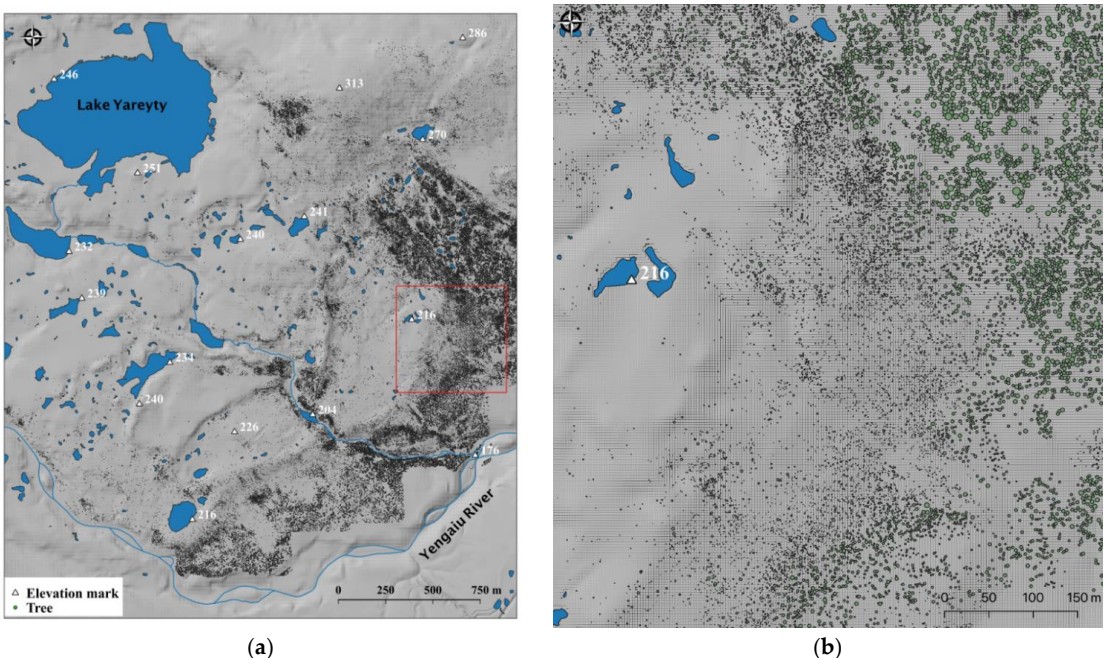

**Figure 4.** Distribution map for the specimens of Siberian larch in the upper treeline ecotone, developed by deciphering aerial images captured with a drone (**a**), and an enlarged fragment (shown on Figure 4a with a red rectangle (**b**)).

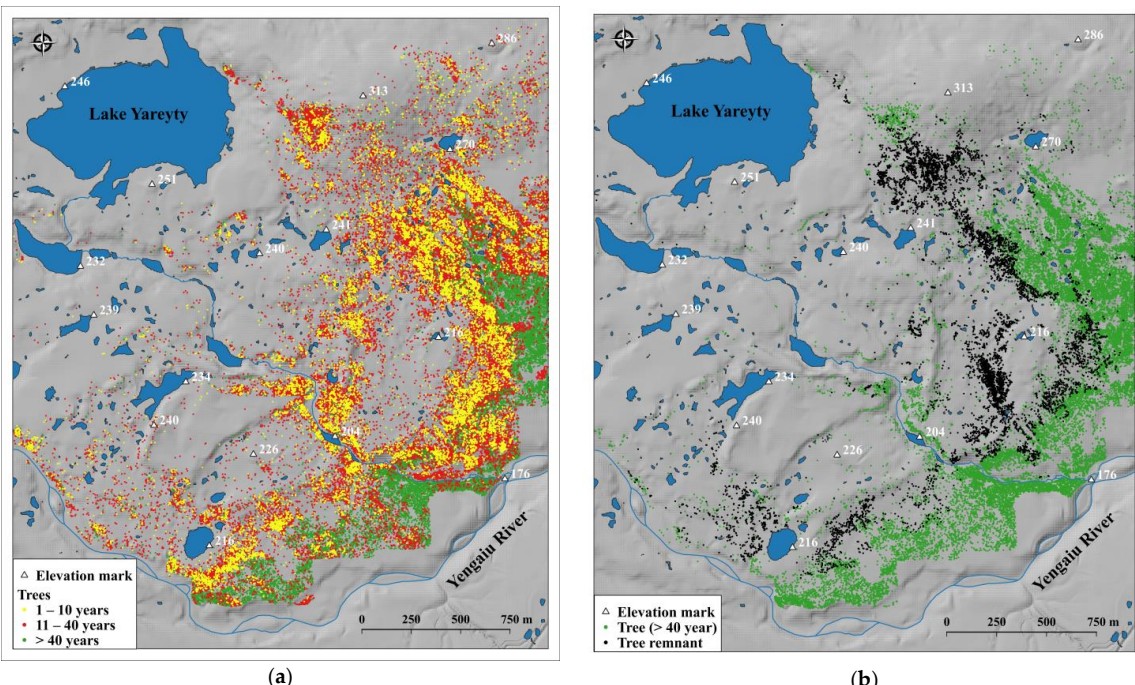

**Figure 5.** Map of Siberian larch tree distribution into three age categories: 1–10, 11–40, and 40+ years (**a**), and the location of trees that were previously growing in the region and died during the Little Ice Age, and trees more than 40 years of age (**b**).

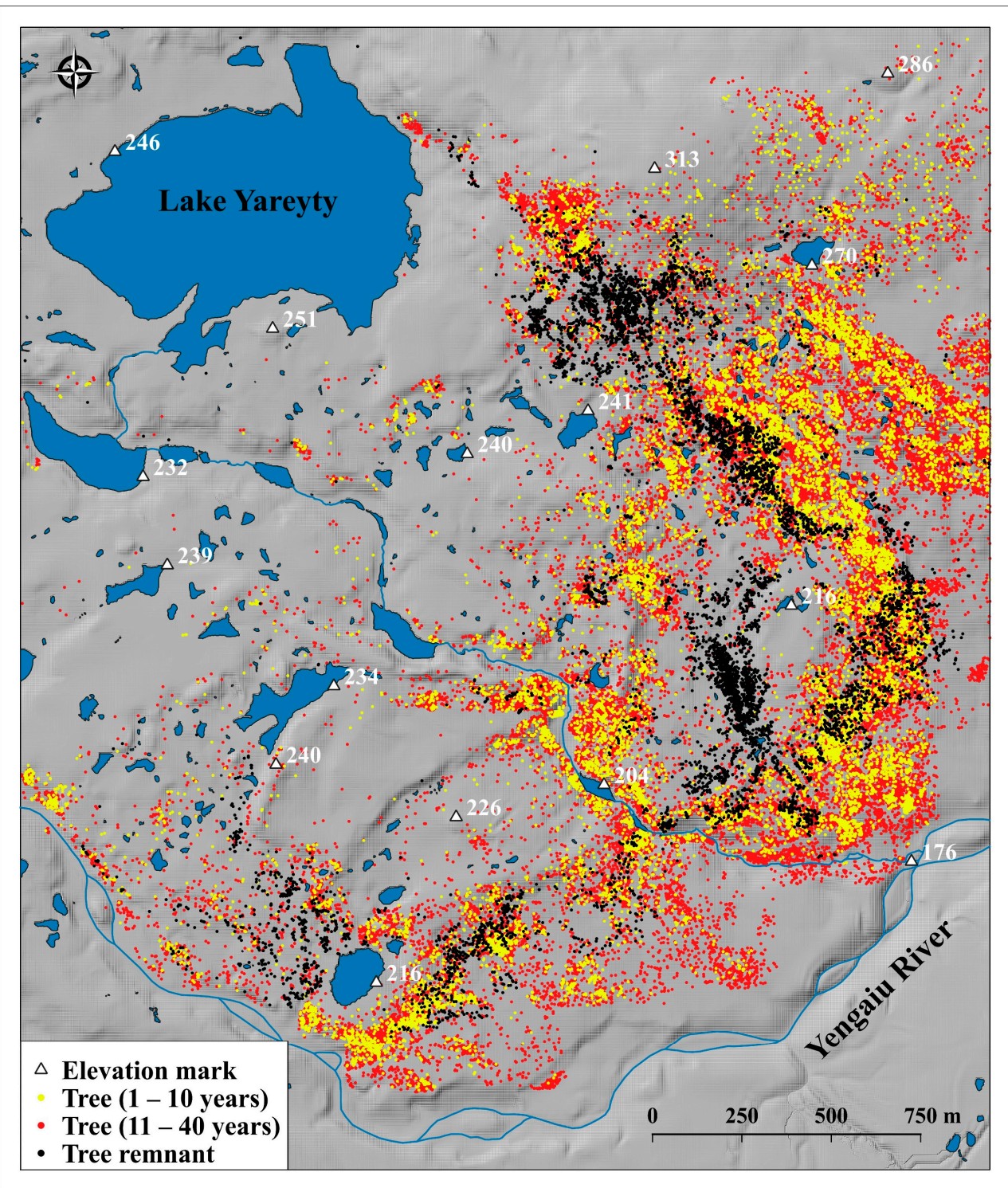

**Figure 6.** Map of Siberian larch trees belonging to two age categories: 1–10 and 11–40 years of age, showing the locations of trees that were growing in the study area and died during the Little Ice Age.

## 4. Discussion

Selection of Siberian larch generations using crown radii allows reconstruction of the tree proliferation process into tundra during the course of modern climate warming. Maps showing the location of Siberian larch generations, and remnants of the trees that grew in the region and died during the Little Ice Age, allow for studying the laws of tree advancement into the previously unforested or sparsely forested parts of the study area.

Some tree remnants were dated as far back as AD 720 [27]. Maximum density of the trees estimated by the number of tree remnants (50–60 trees/ha) was observed from the end of the 12th century to the second half of the 13th century. After 1280–1300, the number of trees started declining, reaching a minimum between AD 1360 and 1450. The next period of tree abundance occurred from AD 1660 to 1740. The next decline started after 1730. It was found that the second minimum number of trees occurred in the second half of the 19th century [27,28]. The trends described were confirmed by dendrochronological dating of tree remnants in a high-elevation transect located 5 km northeast from the upper-right corner of the study area [29].

Figure 5a demonstrates that the maximum number of larch specimens from the 1–10 year age category (yellow dots) is surrounded by trees from the 11–40 and 40+ age categories. This pattern can be due to several reasons, including presence of the trees in fruiting stage that serve as a semination source for the territory, and protection of younger trees from snow abrasion by larger trees [8,17,27]. The map presented in Figure 5b is based on a recent satellite image [17] and supports a previous conclusion that mature modern trees did not yet reach some parts of the study area where trees grew previously. Spatial resolution for drone images is greater than the resolution for previously used satellite images, and it was impossible to recognize smaller trees from younger generations of Siberian larch on satellite images.

It is necessary to note certain specifics of recognizing trees on drone images that can affect location determination of young larch specimens. Tree crowns can block young specimens from an observer. Usually, the sites with large mature trees (lower part of the treeline ecotone) have a well-developed live ground cover including a shrub layer composed of dwarf birch (*Betula nana* L.), lichens, and green mosses. Natural recovery of larch trees in these areas is hampered because seeds cannot obtain access to the mineralized soil layer, while seedlings are subject to severe competitive impact from other plant species occupying lower vegetation layers.

Results of processing drone images and data analysis demonstrate that individual Siberian larch trees belonging to the 1–10 and 11–40 year age intervals already occupied multiple habitats with no tree remnants or a small number of remnants (Figure 6). Densities of modern trees did not yet reach density values that existed in the past on two sites: around the elevation mark of 313 m, and between 204 and 216 m.

Locations of tree remnants indicate parts of the study area that were favorable for the growth of woody vegetation in the past. However, that does not mean that similar favorable conditions currently exist in the same spots. For example, there are a relatively small number of modern trees on-site with numerous tree remnants in the area between the elevation marks of 216 and 204 m. This is due to the fact that in the past, after death of the trees, upper horizons of the soil were eroded by temporary watercourses, exposing many rocks with no fine soil in between (as it was eroded annually). Competition from a well-developed shrub layer of dwarf birch on some parts of this site also makes it difficult to populate it with larch. We also cannot claim that wood remnants themselves can contribute to the restoration of woody vegetation in places it previously grew.

Convincing evidence for the absence of wildfires and other catastrophic phenomena, as well as lacking evidence of significant anthropogenic impact on woody vegetation in the study area, indicate high probability that forest dynamics in the region can be generally attributed to climate change [27].

Dendrochronological dating of tree remnants cannot indicate the cause of death for particular trees. The assumption on the fact that expansion of woody vegetation into forest tundra and tundra is associated with an increase in air temperature is confirmed by results obtained during dendrochronological studies [27,28] and radiocarbon-dating of woody remnants of Scots pine (*Pinus sylvestris* L.) in the northern Swedish Scandes [30].

An earlier start of the growing season is essential for the expansion of woody vegetation into the mountain tundra. For example, the temperature increase in May from −2.4 °C in 1883–1920 to −0.1 °C in 2005–2014 led to an increase in duration of the growing

season by 5–7 days [29]. The average temperature for July from the beginning of the 1880s to the end of the 1990s increased from 13.8 °C to 14.3 °C, and the average temperature of June and July, which is critical for tree growth, increased by 0.9 °C [31]. The mean annual temperature from the beginning of 20th to the beginning of the 21st century increased by about 3 °C [17].

Many researchers note the fact that increase in forest area in the upper parts of the mountain slopes and the upward shift of the treeline coincided with an increase in winter precipitation during the 20th century [8,28,32,33]. According to data from the Salekhard weather station, located 50 km east of the research area, the amount of precipitation in summer months did not change in 2005–2014 compared to 1921–2004, and in winter, it increased by 8 mm [29]. The annual precipitation from the beginning of the 20th to the beginning of the 21st centuries increased by about 200 mm [17]. A positive correlation has been established between the canopy cover, snow depth, and soil temperature, which indicates that increasing canopy cover contributes to snow accumulation and, consequently, to more favorable microclimate [8,29].

However, it should be noted that increased precipitation in winter, affecting the accumulation of snow, can have both positive and negative effects on woody plants. Deep snow cover can significantly delay the beginning of the growing season [2]. On the windswept upper parts of hills and ridges, where the snow depth is low, young larch trees suffer greatly from snow abrasion. The probability of their survival increases if they are protected from wind by large single trees or groups of trees, as well as microrelief features such as large stones or small ridges. The positive effect of snow accumulation is that it can protect a younger generation of trees from frost and snow abrasion [17].

The issue quantitatively assessing younger larch generation survivability requires further studies. Recent years demonstrate both positive impact of local climate warming, and increase in previously insubstantial impacts of biotic factors, including effects of phytopathogens (larch needle blight) and anthropogenic impacts, primarily related to the increase of anthropogenic impact due to reindeer grazing [34].

Allometric relationships between a tree crown diameter and other biometric characteristics of the tree are widely used in forest assessment, including areas close to the upper limit of woody vegetation growth [35,36]. Allometric models for estimating the age of a tree by its crown diameter, without other biometric parameters, are usually characterized by low quantitative adequacy [37]. Determination of tree age categories in the upper treeline ecotone can be performed based on tree height [36]. In this approach, however, it is necessary to consider specifics of meso- and microclimatic conditions in a particular study area. That includes, for example, parameters of snow accumulation, which can significantly affect the growth rate of trees, especially at early stages of ontogenesis. This means that when using tree height, it is necessary to perform justification (based on prior data or probabilistic assessment) for specific tree height thresholds that were chosen to assess tree age. From this perspective, the probabilistic approach proposed in our paper presents a high level of objectivity and universality.

## 5. Conclusions

Vegetation at the upper tree line is a sensitive indicator for changes in regional climate. The method was developed for selection of three age generations of Siberian larch (1–10, 11–40, and 40+ years) growing in the upper tree line ecotone of the Rai-Iz massif (Polar Urals, Russia). The method uses probability statistical models based on average crown radius. A map demonstrating location of individual larch trees (88,833 specimens) and the size of their crowns in the research area was created based on their recognition in aerial images captured from 50 m altitude with an unmanned aerial vehicle on a 7.32 square kilometer territory. The layer was used to draw maps showing the three age categories of Siberian larch, and displaying data previously produced on the location of trees that were growing within the study area and died during the Little Ice Age (from the end of the 13th century through the end of the 19th century). That allowed assessment of larch

proliferation into the upper part of an ecotone occurring due to climate warming observed in the Polar Urals in the 20th and early 21st century. It was found that larch exemplars from the 1–10 and 11–40 years of age intervals have already occupied habitats where there are few or no old tree remnants. The method suggested for reconstruction of Siberian larch expansion in the upper tree line ecotone can be applied for reconstructing climate-driven spatiotemporal dynamics of tree vegetation at upper growth boundaries in other regions.

**Author Contributions:** Conceptualization: V.F. and A.M.; formal analysis, V.F.; methodology, V.F. and A.M.; data curation, V.F., A.M., D.G. and E.A.; software, V.F.; visualization, V.F.; project administration, V.F. and A.M.; writing—original draft, V.F.; writing—review and editing, A.M. All authors have read and agreed to the published version of the manuscript.

**Funding:** This research was collaboratively funded by the Russian Ministry for Science and Education (projects No. FEUG-2020-0013, No. FEUG-2021-0009). An approach to the selection of generations of trees by crown diameter was developed within the framework of the project of the Russian Ministry for Science and Education No. FEUZ-2021-0014.

**Data Availability Statement:** The data are given in the article.

**Conflicts of Interest:** The authors declare no conflict of interest.

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
