# Peer review of "Reconstruction of the Expansion of Siberian Larch into the Mountain Tundra in the Polar Urals in the 20th—Early 21st Centuries"

_forests, doi:10.3390/f13030419_

Round 1

Reviewer 1 Report

Substantial revisions were made in the revised manuscript. The authors answered all the question I asked. Using aerial images and field data, they reconstruct the spatial distributions of age structure for Siberian larch treeline ecotone. The idea is very good. The results are interesting to readers.

Author Response

Dear reviewer!

Thank you for your informative comments and suggestions for improving our article!

with best regards,

Valery Fomin

Reviewer 2 Report

The manuscript "Reconstruction of the expansion of Siberian larch into the mountain tundra in the Polar Urals in the 20th – early 21th centuries” is a revised version of a manuscript submitted to MDPI Forests before, which I was asked to review a couple of weeks ago. After careful examination I opted for reject back then, due to several shortcomings, but encouraged a resubmission after a thorough revision. The paper deals with spatio-temporal changes in the distribution of larch specimens within and beyond elevational treeline ecotone of the Polar Urals during the last seven decades, employing high resolution aerial images made by an unmanned aerial vehicle (UAV). An upward shift of the treeline is a common feature in many mountain areas of the worlds, the reasons for such shifts, however, vary from case study to case study (e.g. rising carbon dioxide and nitrogen concentration in the atmosphere, warming temperatures, modified precipitation patterns, changes in anthropogenic land use), so considering the still existing uncertainties about treeline shifts, further studies on this topic are useful.

Many of my original concerns about this manuscript are reasonably addressed in the revised version, but there are still some points which should be modified and corrected by the authors in a moderate revision. As this was not an available choice, I opted for major revision. In addition, despite not being a native English speaker, I still believe that the manuscript requires a thorough linguistic improvement by a native English speaker to erase bad wording and grammatical errors.

  • A key aspect of the manuscript is the application of UAV imagery for the study of treeline shifts. This could make a valuable contribution to a broad audience; however, I would like to get some more information on the procedure of the drone flights and the creation of orthophotos by the UAV imagery in the Material and Methods section.
  • L70: “… ending et about 10 Kya”; Please also provide the denomination of this cold period: is it Waldai?
  • L76-79: This is confusing: in L76-77 you write that the study area is in the permafrost zone and in L78-79 you state that there is no permafrost at the study sites. Please clarify!
  • The authors throughout the manuscript mix the terms height, elevation and altitude. In L80 for example the term height is used, but what the authors mean is elevation. In the figure caption of Figure 1 altitudinal benchmark should be replaced by elevational benchmark: in L121: altitude is the right term, not height! Please carefully check the whole manuscript for the right and consistent use of the terms according to the definitions given in McVicar, Tim & Körner, Christian. (2012): On the use of elevation, altitude, and height in the ecological and climatological literature. Oecologia. 171. 10.1007/s00442-012-2416-7.
  • L104: shoulders of the increment borer: I guess what you mean is the height of the shoulder of the person who is using the increment borer. Please reword
  • L106-107: Please provide a reference for this correcting procedure
  • L111: shaded relief instead of shaded model
  • L119: sample site instead of test site; please check the whole ms, there might be more “test sites”.
  • L184: site is not the right term in this context, it is either age category I, II, III or age class I, II, III; check and replace throughout the manuscript
  • L189-196: Figure caption of Figure 3 is redundant to L179-187; remove text at either location
  • L203: Little Ice Age is not “recent”, it ended more than 150 years ago. I recommend to remove the word recent
  • Legends of Figure 5 and 6 are hard to read: Please modify the legend and enlarge the symbols
  • L213-214: Use upper case letters: Little Ice Age

Author Response

Dear reviewer!

The answers to your comments and suggestions are given in the attached file.

Thank you!

with best regards,

Valery Fomin

Round 2

Reviewer 2 Report

Dear authors,

thank you for your efforts! All of my earlier concerns are well addressed by the authors in the revised version and I believe the manuscript significantly improved. I recommend with two very small additional modifications:

L71: I recommend not to use the word "occurred" in this context, as the last global glaciation lasted for many tens of thousands of years. This cold phase ended at about 10.000 years b.P. So instead of occurred I believe ended is the more correct word.

L242: change high altitude-transect to high-elevation transect  

Author Response

Dear reviewer,

We agree with your comments and have made changes to the article in accordance with your recommendations. We are impressed by your professional work to improve our manuscript. Thank you very much!

With best regards
Valery Fomin
